# Design and Synthesis of HCV-E2 Glycoprotein Epitope Mimics in Molecular Construction of Potential Synthetic Vaccines

**DOI:** 10.3390/v13020326

**Published:** 2021-02-20

**Authors:** Theodorus J. Meuleman, Vanessa M. Cowton, Arvind H. Patel, Rob M. J. Liskamp

**Affiliations:** 1School of Chemistry, University of Glasgow, Joseph Black Building, University Avenue, Glasgow G12 8QQ, UK; th.j.meuleman@gmail.com; 2Enzytag, Daelderweg, 9 6361 HK Nuth, The Netherlands; 3MRC-University of Glasgow Centre for Virus Research, Garscube Campus, Sir Michael Stoker Building, 464 Bearsden Road, Glasgow G61 1QH, UK; vanessa.cowton@glasgow.ac.uk; 4Department of Biochemistry, Cardiovascular Research Institute Maastricht (CARIM), Faculty of Health, Medicine and Life Sciences, Maastricht UMC, Universiteitssingel 50, 6229 ER Maastricht, The Netherlands; 5Chemical Biology and Drug Discovery, Department of Pharmaceutics, Utrecht University, Universiteitsweg 99, 3584 CG Utrecht, The Netherlands; 6Cristal Therapeutics, Oxfordlaan 55, 6229 EV Maastricht, The Netherlands

**Keywords:** scaffolding, synthetic vaccine, protein mimic, cyclic peptide, click reaction, epitope mimic, envelope glycoprotein

## Abstract

Hepatitis C virus remains a global threat, despite the availability of highly effective direct-acting antiviral (DAA) drugs. With thousands of new infections annually, the need for a prophylactic vaccine is evident. However, traditional vaccine design has been unable to provide effective vaccines so far. Therefore, alternative strategies need to be investigated. In this work, a chemistry-based approach is explored towards fully synthetic peptide-based vaccines using epitope mimicry, by focusing on highly effective and conserved amino acid sequences in HCV, which, upon antibody binding, inhibit its bio-activity. Continuous and discontinuous epitope mimics were both chemically synthesized based on the HCV-E2 glycoprotein while using designed fully synthetic cyclic peptides. These cyclic epitope mimics were assembled on an orthogonally protected scaffold. The scaffolded epitope mimics have been assessed in immunization experiments to investigate the elicitation of anti-HCV-E2 glycoprotein antibodies. The neutralizing potential of the elicited antibodies was investigated, representing a first step in employing chemically synthesized epitope mimics as a novel strategy towards vaccine design.

## 1. Introduction

Vaccination has been an essential and successful approach in controlling a wide variety of virus infections throughout history [1,2]. However, the emergence of more complex viruses, like the human immunodeficiency virus (HIV) [3,4] and hepatitis C virus (HCV) [5,6,7], has not enjoyed similar success. Traditional approaches for vaccine design, including inactivated, attenuated, subunit, and recombinant vaccine strategies, appeared to be unable to cope with the level of complexity that is associated with these viruses. The major obstacles for effective vaccine design against, for example, HCV, can be attributed to its high mutation rate that results in viral escape [8,9]. The immune system struggles to adapt to these ever-changing viruses and, therefore, unable to resolve the infection naturally in most of the cases [10]. Furthermore, the complexity of these viruses can be found in their adopted strategies to negatively influence the immune system to maintain infection. For HCV, this includes the manipulation of communication within the immune system [10,11], as well as providing a wide variety of shielding and decoy factors (i.e., glycan shielding [12], association with host lipoprotein [13], and immunodominant epitopes [14]), which hinder or occupy the immune system without reducing the infectivity and subsequent biological effects of the virus.

Instead of only looking towards previously developed strategies to target these new threats that are posed by these viruses, it may be necessary to adopt alternative strategies to develop effective vaccines. One such strategy could be found in mimicry of crucial and exposed viral proteins [15,16]. Conceptually, protein mimicry is based on capitalizing on known peptide sequences (epitopes) within the viral proteins that reduce virus efficacy when targeted and recognized by the antibodies of the immune system. These epitopes can be synthesized by solid phase peptide synthesis (SPPS) [17,18] and presented as a (synthetic) vaccine to induce a more targeted immune response devoid of any immunomodulatory effects that are inherent to the intact virus [19,20]. In addition, these epitopes can be identified to be highly conserved and resistant to escape mutations that result in decreased efficacy [21,22,23,24,25,26,27]. Even though it is unlikely to have absolute conservation of the epitope or to completely rule out the possibility of escape mutations, the synthetic approach allows for a rapid modular approach that can quickly adapt to viral variation by simply exchanging the synthetic peptides. Thereby, it can provide a tool to rapidly respond to the dynamic and changeable nature of viruses, like HIV and HCV.

However, successful mimicry of peptide epitopes does not only depend on synthetic peptides with the correct amino acid sequence. Instead, these epitopes often have complex spatial conformations when present within the viral protein that need to be included in a synthetic vaccine [15,16]. Such conformations can include loops, α-helices, and β-sheet-like structures. Epitopes can be targeted as one single continuous sequence of amino acids, referred to as a continuous epitope. Alternatively, discontinuous epitopes consist of multiple peptide segments within the viral protein that form a recognition site by the overall folding of the protein. Consequently, these peptide sequences can be far removed from each other within the primary structure of the viral protein. Therefore, mimicry of a discontinuous epitope is significantly more challenging. Whereas a continuous epitope might be successfully mimicked by a single synthetic linear or cyclic peptide, a discontinuous epitope requires the incorporation of multiple different synthetic peptides into the same vaccine construct. Ideally, these multiple synthetic peptides must be incorporated into a single molecular structure that is capable of preserving their original spatial orientation with respect to each other, as was present in the viral protein. Such single molecular structures that are capable of carrying multiple different (cyclic) peptide segments are referred to as molecular scaffolds. It is important to realize that, despite consisting of a single peptide segment, even continuous epitope mimics must generally be constrained, for example, by cyclization, to induce (a) similar conformation(s) otherwise induced by the intact (viral) protein structure to be mimicked.

Our group has extensively investigated approaches to assemble multiple synthetic peptides onto one individual (scaffold)molecule towards development of synthetic receptors [28], antibodies [29,30], and vaccines [19,20,31]. With respect to this, Werkhoven et al. [20,31] developed an orthogonally protected trialkyne triazacyclophane (TAC) scaffold that can incorporate different synthetic peptides by copper(I) catalyzed alkyne-azide cycloaddition (CuAAC) and has used it towards the development of synthetic vaccines. In this work, the TAC-scaffold that was developed by Werkhoven et al. [31] was also improved by introducing a suitably protected linker for attachment to a carrier protein that was compatible with CuAAC.

Previous work involved promising epitopes within the HCV-E2 glycoprotein that were chemically synthesized and evaluated *in vitro* as continuous epitope mimics [32,33]. The present work described here reports on the selective attachment of these cyclic peptides to the developed TAC-scaffold to obtain a mimic of the discontinuous epitope. After molecular construction, both continuous and discontinuous epitope mimics were conjugated to a mariculture keyhole limpet hemocyanin (mcKLH) carrier protein. The conjugated epitope mimics were used in an immunization experiment to elicit anti-HCV E2 glycoprotein antibodies to evaluate their potential as synthetic vaccines.

## 2. Experimental Methods

### 2.1. General Procedure

All of the reagents and solvents were used as received. Fmoc-amino acids were obtained from Activotec (Cambridge, Comberton, United Kingdom) and *N*,*N*,*N*′,*N*′-Tetramethyl-*O*-(6-chloro-1*H*-benzotriazol-1-yl)uranium hexafluorophosphate (HCTU) was obtained from Matrix Innovation (Quebec, Canada). Tentagel S RAM resin (particle size 90 μm, capacity 0.25 mmol·g^−1^) was obtained from IRIS Biotech (Marktredwitz, Germany). Methyl *tert-*butyl ether (MTBE), Hexane (HPLC grade), and trifluoro acetic acid (TFA) were obtained from Aldrich (Milwaukee, USA). DMF (Peptide grade) was obtained from VWR (Lutterworth, UK). Piperidine, D*i*PEA were obtained from AGTC Bioproducts (Hessle, UK) and 1,2-ethanedithiol (EDT) was obtained from Merck (Darmstadt, Germany). HPLC grade CH_2_Cl_2_ and acetonitrile were obtained from Fischer Scientific (Loughborough, UK). Solid phase peptide synthesis was performed on a PTI Tribute-UV peptide synthesizer (Gyros Protein Technologies, Uppsala, Sweden). Lyophilizations were performed on a Christ Alpha 2–4 LD*plus* apparatus (Martin Christ Gefriertrocknungsanlagen GmbH (Osterode am Harz, Germany). The reactions were carried out at ambient temperature, unless stated otherwise. The solvents were evaporated under reduced pressure at 40 °C. Reactions in solution were monitored by TLC analysis and R_f_-values were determined on Merck pre-coated silica gel 60 F-254 (0.25 mm) plates. The spots were visualized by UV-light and permanganate stain. Column chromatography was performed on Silicaflash P60 (40–63 μm) from Silicycle (Quebec City, Canada) or on a Biotage Isolera One purification system while using prepacked silica (KP-SIL) Biotage (Uppsala, Sweden) SNAP cartridges. ^1^H NMR data was acquired on a Bruker 400 MHz spectrometer in CDCl_3_ as solvent. Chemical shifts (δ) are reported in parts per million (ppm) relative to trimethylsilane (TMS, 0.00 ppm). Analytical high-pressure liquid chromatography (HPLC) was carried out on a Shimadzu instrument (Kioto, Japan) comprising a communication module (CBM-20A), autosampler (SIL-20HT), pump modules (LC-20AT), UV/Vis detector (SPD-20A), and system controller (Labsolutions V5.54 SP), with a Phenomenex (Torrance, CA, USA) Gemini C18 column (110 Å, 5 μm, 250 × 4.60 mm) or Dr. Maisch (Ammerbuch, Germany) Reprosil Gold 200 C18 (5 μm, 250 × 4.60 mm). UV measurements were recorded at 214 and 254 nm, while using a standard protocol: 100% buffer A (acetonitrile/H_2_O 5:95 with 0.1% TFA) for 2 min. followed by a linear gradient of buffer B (acetonitrile/H_2_O 95:5 with 0.1% TFA) into buffer A (0–100% or 0–50%) over 30 min. at a flow rate of 1.0 mL·min^−1^. Liquid chromatography mass spectrometry (LCMS) was carried out on a Thermo Scientific LCQ Fleet quadrupole mass spectrometer with a Dionex (Thermo Scientific, Waltham, MA, USA) Ultimate 3000 LC using a Dr. Maisch Reprosil Gold 120 C18 column (110 Å, 3 μm, 150 × 4.0 mm) and the same linear gradients of buffer B into buffer A, flowrate, and buffers, as described for the analytical HPLC. Purification of the peptidic compounds was performed on an Agilent Technologies 1260 infinity preparative system using both UV and ELSD (Agilent, Santa Clara, CA, USA) detectors with a Dr. Maisch Reprosil Gold 200 C18 (10 μm, 250 × 20 mm). The auto-collection of fractions was used based on the UV measurements at 214 nm, utilizing customized protocols using the same buffers described for the analytical HPLC.

### 2.2. General Method for Automated Peptide Synthesis

The peptides were synthesized on a PTI Tribute-UV peptide synthesizer. Tentagel S RAM resin (1.0 g, 0.25 mmol, 1.0 equiv. or 0.43 g, 0.10 mmol, 1.0 equiv.) was allowed to swell in DMF (3 × 10 min). De-protection of the Fmoc group was achieved by treatment of the resin with 20% piperidine in DMF using the RV_top_UV_Xtend protocol from the Tribute-UV peptide synthesizer, followed by a DMF washing step (5 × 30 s). Fmoc-protected amino acids were coupled using HCTU (on the 0.10 mmol scale 5.0 equiv. were used and on the 0.25 mmol scale 4.0 equiv. were used) and D*i*PEA (on the 0.10 mmol scale 10 equiv. were used and on the 0.25 mmol scale 8.0 equiv. were used) in DMF, as a coupling reagent, with 2 min. pre-activation. The coupling time was 10 min. when the peptide was synthesized on a 0.10 mmol scale and 20 min. when the 0.25 mmol scale was conducted. After every coupling, the resin was washed with DMF (6 × 30 s), followed by a capping step for 10 min. using capping solution (24 mL acetic anhydride and 11 mL D*i*PEA in 500 mL DMF; using 3 mL for the 0.10 mmol scale and 6 mL for the 0.25 mmol scale). After coupling of the last amino acid, the Fmoc group was deprotected using the standard de-protection conditions (described above) and the resulting free N-terminus was maintained (peptide **1**) or acetylated (peptides **2** and **3**) by treating the resin bound peptide with capping solution (24 mL acetic anhydride and 11 mL D*i*PEA in 500 mL DMF; using 3 mL for the 0.10 mmol scale and 6 mL for the 0.25 mmol scale) for 10 min. After the last step, the resin was washed with DMF (5 × 30 s) and then dried over a nitrogen flow for 10 min. Cleavage and de-protection was achieved by treatment of the resin with TFA/H_2_O/TIS/EDT (15 mL for the 0.25 mmol scale and 5 mL for the 0.10 mmol scale, 90/5/2.5/2.5, *v*/*v*/*v*/*v*) for 3 h at room temperature. The peptide was then precipitated with Et_2_O (90 mL for the 0.25 mmol scale and 45 mL for the 0.10 mmol scale), centrifuged (4500 rpm; 5 min.), the supernatant decanted, and the pellet washed three times with Et_2_O (90 mL for the 0.25 mmol scale and 45 mL for the 0.10 mmol scale). The resulting pellet was re-dissolved in *t*BuOH/H_2_O (1/1, *v/v*) and lyophilized.

Peptide 1. This peptide was synthesized, as described above, by solid phase synthesis on a 0.25 mmol scale, which afforded crude peptide (0.28 g). This crude peptide was used in the cyclization by alkylation step. *t*_R_ = 16.5 min.; Low-Resolution(LR)MS: *m/z* calculated for C_79_H_127_N_27_O_22_S_2_: 935.96 ½[*M*+2H]^2+^; found: 936.75.

Peptide 2. This peptide was synthesized, as described above, by solid phase synthesis on a 0.25 mmol scale, which afforded crude peptide (0.35 mg). This crude peptide was used in the cyclization by an alkylation step. *t*_R_ = 19.2 min.; LRMS: *m/z* calculated for C_89_H_117_N_21_O_19_S_2_: 924.92 ½[*M*+2H]^2+^; found: 925.67.

Peptide 3. This peptide was synthesized, as described above, by solid phase synthesis on a 0.25 mmol scale, which afforded crude peptide (0.35 g). This crude peptide was used in the cyclization by alkylation step. *t*_R_ = 16.7 min.; LRMS: *m/z* calculated for C_96_H_140_N_28_O_31_S_2_: 1123.49 ½[*M*+2H]^2+^; found: 1124.17.

### 2.3. General Method for Peptide Cyclization

Peptide cyclization was carried out, as described by Meuleman et al. [33].

Cyclic peptide **6**. Crude peptide **3** (0.35 g, 0.25 μmol, 1.0 equiv.) was treated with azido triazinane-tris(2-bromoethanone (TADB-N_3_) [34] (0.12 g, 0.30 mmol, 1.2 equiv.). Purification was done in batches of 30–40 mg crude dissolved in 3 mL buffer A:B (75:25, *v/v*) while using a custom protocol: 27% buffer B in buffer A for 2 min., followed by a linear gradient of 27–38% buffer B in buffer A for 10 min., which afforded cyclic peptide 6 (0.12 g, 49 μmol, 20%). t_R_ = 17.4 min.; High Resolution(HR)MS: calculated *m/z* for C_105_H_150_N_34_O_34_S_2_: 1248.5326 ½[*M*+2H]^2+^; found 1248.5361; LRMS: calculated *m/z* for C_105_H_150_N_34_O_34_S_2_: 1248.53 ½[*M*+2H]^2+^; found 1249.58.

Tetraethylene glycol monotrityl thioether (**7**). Synthesized, as described Meuleman et al. [32].

Tetraethylene glycol monotrityl thioether p-toluenesulfonate (**9**). Tetraethylene glycol monotrityl thioether **7** (2.6 g, 5.8 mmol, 1.0 equiv.) was dissolved in anhydrous CH_2_Cl_2_ (50 mL). The resulting solution was cooled to 0 °C using an ice bath, followed by the addition of TEA (1.6 mL, 12 mmol, 2.1 equiv.). After 1 h, p-toluenesulfonyl chloride 8 (2.3 g, 12 mmol, 2.1 equiv.) was added. The resulting reaction mixture was stirred overnight at room temperature. Upon completion of the reaction, as determined by TLC (50% EtOAc in petroleum ether 40–60 °C), the mixture was washed with H_2_O (100 mL). Next, the aqueous layer was back-extracted with CH_2_Cl_2_ (2 × 100 mL). Subsequently, the combined organic layers were washed using brine (1 × 100 mL), dried over MgSO_4_, and then filtered. The filtrate was concentrated *in vacuo* and purified by automated flash column chromatography using a linear gradient (30–50% EtOAc in petroleum ether 40–60 °C over 10 column volumes). Pure fractions were combined and concentrated *in vacuo*, affording pure tetraethylene glycol monotrityl thioether ρ-toluenesulfonate 9 (2.6 g, 4.3 mmol, 74%) as a yellowish oil. R_f_ = 0.52 (50% EtOAc in petroleum ether 40–60 °C); t_R_ = 46.6 min.; ^1^H-NMR (400 MHz, CDCl_3_): δ = 7.79 (d, ^3^J_HH_ = 8.2 Hz, 2H, aryl *o*-*H*), 7.41 (m, 6H, trityl *o*-*H*), 7.32 (d, ^3^J_HH_ = 8.2 Hz, 2H, aryl *m*-*H*), 7.28 (m, 6H, trityl *m*-*H*), 7.21 (m, 3H, trityl *p*-*H*), 4.14 (t, ^3^J_HH_ = 4.8 Hz, 2H, TsOC*H*_2_), 3.66 (t, ^3^J_HH_ = 4.8 Hz, 2H, TsOCH_2_C*H*_2_), 3.53 (m, 6H, 3 × C*H*_2_), 3.43 (m, 2H, C*H*_2_), 3.30 (t, ^3^J_HH_ = 6.9 Hz, 2H, SCH_2_C*H*_2_), 2.43 (t, ^3^J_HH_ = 6.9 Hz, 2H, SC*H*_2_), 2.43 (s, 3H, C*H*_3_) ppm; ^13^C-NMR (101 MHz, CDCl_3_): δ = 144.8 (aryl-*C*), 144.7 (aryl-*C*), 133.1 (aryl-*C*), 130.1 (aryl-*C*), 129.8 (aryl *m*-*C*H), 129.6 (trityl *o*-*C*H), 128.0 (aryl *o*-*C*H), 127.9 (trityl *m*-*C*H), 126.6 (trityl *p*-*C*H), 70.7 (*C*H_2_), 70.5 (*C*H_2_), 70.5 (*C*H_2_), 70.2 (*C*H_2_), 69.6 (SCH_2_*C*H_2_), 69.2 (TsO*C*H_2_), 68.7 (TsOCH_2_*C*H_2_), 66.6 (*C*S), 31.7 (S*C*H_2_), 21.6 (*C*H_3_) ppm; HRMS: calculated *m/z* for C_34_H_38_O_6_S_2_: 629.2008 [*M*+Na]^1+^; found 629.1996.

Triethylene glycol monotrityl thioether ethylamine (**10**). p-toluenesulfonate tetraethylene glycol monotrityl thioether 9 (2.6 g, 4.3 mmol, 1.0 equiv.) was dissolved in MeCN (100 mL), followed by the addition of aqueous 25% ammonia (50 mL). The resulting reaction mixture was stirred overnight at room temperature. Although the reaction was not complete according to TLC (20% MeOH in CH_2_Cl_2_), work-up was carried by the removal of MeCN *in vacuo* and extraction of the aqueous layer with CH_2_Cl_2_ that was supplemented with 1% TEA (3 × 50 mL). The combined CH_2_Cl_2_ layer were washed with brine (100 mL), dried over NaSO_4_, and then filtered. The filtrate was concentrated *in vacuo*, followed by purification using automated flash column chromatography using a linear gradient (0–10% MeOH in CH_2_Cl_2_ supplemented with 1% TEA over 10 column volumes). Pure triethylene glycol monotrityl thioether ethylamine 10 (0.26 g, 0.58 mmol, 14%) was obtained as a yellowish oil. R_f_ = 0.27 (10% MeOH in CH_2_Cl_2_ supplemented with TEA); t_R_ = 34.4 min.; ^1^H-NMR (400 MHz, CDCl_3_): δ = 7.41 (m, 6H, trityl *o*-*H*), 7.27 (m, 6H, trityl *m*-*H*), 7.21 (m, 3H, trityl *p*-*H*), 3.61 (m, 4H, 2 × C*H*_2_), 3.57 (m, 2H, C*H*_2_), 3.50 (t, ^3^J_HH_ = 5.2 Hz, 2H, C*H*_2_CH_2_NH_3_^+^), 3.45 (m, 2H, C*H*_2_), 3.31 (t, ^3^J_HH_ = 6.9 Hz, 2H, SCH_2_C*H*_2_), 2.85 (t, ^3^J_HH_ = 5.2 Hz, 2H, C*H*_2_NH_3_^+^), 2.43 (t, ^3^J_HH_ = 6.9 Hz, 2H, SC*H*_2_), 1.91 (broad s, 3H, NH_3_^+^) ppm; ^13^C-NMR (101 MHz, CDCl_3_): δ = 144.8 (trityl-*C*), 129.6 (trityl *o*-*C*H), 127.9 (trityl *m*-*C*H), 126.7 (trityl *p*-*C*H), 73.1 (*C*H_2_CH_2_NH_3_^+^), 70.6 (*C*H_2_), 70.5 (*C*H_2_), 70.3 (*C*H_2_), 70.2 (*C*H_2_), 69.6 (SCH_2_*C*H_2_), 66.6 (*C*S), 41.7 (*C*H_2_NH_3_^+^), and 31.7 (S*C*H_2_) ppm; HRMS: calculated *m/z* for C_27_H_33_NO_3_S: 452.2259 [*M*+H]^1+^; found 452.2244.

Orthogonally protected trialkyne TAC-scaffold (**11**). This compound was synthesized according to the earlier reported procedure of Werkhoven et al. [31].

Triethylene glycol monotrityl thioether ethylamine that was equipped TAC-scaffold (**12**). TAC-scaffold **11** (42 mg, 53 μmol, 1.0 equiv.) was dissolved in DMF (8 mL). Next, benzotriazol-1-yloxy)tris(dimethylamino)phosphonium hexafluorophosphate (BOP) (30 mg, 68 μmol, 1.3 equiv.) was dissolved in DMF (3 × 2 mL) and subsequently added to the solution containing TAC-scaffold **25**. Subsequently, triethylene glycol monotrityl thioether ethylamine 10 (51 mg, 0.11 mmol) was dissolved in DMF (3 × 2 mL) and added. Finally, D*i*PEA (37 μL, 0.21 mmol, 4.0 equiv.) was added and the reaction mixture was stirred for 1 h at room temperature. Upon completion of the reaction, as determined by LCMS, solvent was removed *in vacuo*. The residue was taken up in *t*BuOH/H_2_O (1/1, *v/v*) and then lyophilized. The crude yellowish oil was dissolved in CH_2_Cl_2_ and purified manually using silica gel chromatography (2% MeOH in CH_2_Cl_2_) and visualized with UV and potassium permanganate stain. Pure fractions were combined and concentrated *in vacuo*, the residue taken up in *t*BuOH/H_2_O (1/1, *v/v*) and lyophilized, affording pure triethylene glycol monotrityl thioether ethylamine equipped TAC-scaffold 12 (52 mg, 43 μmol, 81%) as a white powder. R_f_ = 0.52 (5% MeOH in CH_2_Cl_2_); t_R_ = 49.6 min.; ^1^H-NMR (400 MHz, CDCl_3_): δ = 7.69 (m, 2H, aryl-*H*), 7.39 (m, 6H, trityl *o*-*H*), 7.34 (m, 1H, aryl-*H*), 7.27 (m, 6H, trityl *m*-*H*), 7.20 (m, 3H, trityl *p*-*H*), 6.84 (m, 1H, OCH_2_CH_2_N*H*), 4.61 (m, 4H, 2 × NC*H*_2_-aryl), 3.64 (m, 8H, 4 × C*H*_2_), 3.58 (m, 2H, OCH_2_C*H*_2_NH), 3.43 (m, 6H, NC*H*_2_CH_2_C*H_2_*N and OC*H*_2_CH_2_NH), 3.27 (t, ^3^J_HH_ = 6.8 Hz, 2H, SCH_2_C*H*_2_), 2.96 (m, 4H, NC*H*_2_CH_2_C*H_2_*N), 2.64 (m, 8H, 2 × C*H*_2_C*H*_2_CCSi), 2.44 (m, 4H, C*H_2_*C*H*_2_CCH), 2.39 (t, ^3^J_HH_ = 6.8 Hz, 2H, SC*H*_2_CH_2_), 1.95 (m, 1H, CC*H*), 1.45 (m, 4H, 2 × NCH_2_C*H_2_*CH_2_N), 1.05 (m, 21H, Si(C*H*(C*H*_3_)_2_)_3_), 0.97 (t, ^3^J_HH_ = 7.9 Hz, 9H, Si(CH_2_C*H*_3_)_3_), 0.58 (q, ^3^J_HH_ = 7.9 Hz, 6H, Si(C*H*_2_CH_3_)_3_) ppm; ^13^C-NMR (101 MHz, CDCl_3_): δ = 144.7 (trityl-*C*), 129.6 (trityl *o*-*C*H), 127.9 (trityl *m*-*C*H), 126.8 (aryl-*C*H) 126.6 (trityl *p*-*C*H), 126.2 (aryl-*C*H), 70.6 (*C*H_2_), 70.4 (OCH_2_*C*H_2_NH), 70.3 (*C*H_2_), 70.1 (O*C*H_2_CH_2_NH), 69.7 (*C*H_2_), 69.6 (SCH_2_*C*H_2_), 68.6 (*C*CH), 53.9 (N*C*H_2_-aryl), 51.9 (N*C*H_2_-aryl), 47.8 (N*C*H_2_CH_2_*C*H_2_N), 45.8 (N*C*H_2_CH_2_*C*H_2_N), 45.6 (N*C*H_2_CH_2_*C*H_2_N), 43.5 (N*C*H_2_CH_2_*C*H_2_N), 40.0 (*C*H_2_), 33.0 (*C*H_2_*C*H_2_CCSi), 32.9 (*C*H_2_*C*H_2_CCSi), 31.9 (*C*H_2_*C*H_2_CCH), 31.7 (S*C*H_2_CH_2_), 28.5 (NCH_2_*C*H_2_CH_2_N), 27.5 (NCH_2_*C*H_2_CH_2_N), 18.7 (Si(*C*H(*C*H_3_)_2_)_3_), 18.4 (C*C*H), 16.4 (*C*H_2_*C*H_2_CCSi), 16.4 (*C*H_2_*C*H_2_CCSi), 14.5 (*C*H_2_*C*H_2_CCH), 11.3 (Si(*C*H(*C*H_3_)_2_)_3_), 7.5 (Si(CH_2_*C*H_3_)_3_), and 4.4 (Si(*C*H_2_CH_3_)_3_) ppm; HRMS: calculated *m/z* for C_72_H_100_N_4_O_7_SSi_2_: 1221.6929 [*M*+H]^1+^; found 1221.6870; LRMS: calculated *m/z* for C_72_H_100_N_4_O_7_SSi_2_: 1221.69 [M+H]^1+^/1243.67 [*M*+Na]^1+^; found 1222.92/1244.17.

Tetraethylene glycol p-toluenesulfonate (**13**). This compound was obtained, as described earlier by Meuleman et al. [32].

Tetraethylene glycol Boc-amide (**16**). Tetraethylene glycol p-toluenesulfonate **13** (7.2 g, 21 mmol, 1.0 equiv.) was taken up in aqueous 25% ammonia (50 mL), and the reaction was stirred overnight at room temperature. The solvents were removed *in vacuo* and the residue re-dissolved in MeCN (50 mL) and fresh aqueous 25% ammonia (50 mL) was added because the reaction was not fully complete, as determined by TLC (100% EtOAc) and visualized with a ninhydrin stain. After stirring for an additional 8 h, the reaction was still not complete. Again, the solvents were removed *in vacuo* and the residue was now taken up in aqueous 25% ammonia (50 mL) and stirred overnight at room temperature. Finally, the next morning, the reaction was complete, as determined by TLC (100% EtOAc) and visualized with a ninhydrin stain. The solvent was removed *in vacuo* and the residue was co-evaporated with toluene (2×). Next, the thus obtained crude amine 14 was dissolved in anhydrous THF (50 mL), followed by the addition of di-*tert*-butyl dicarbonate **15** (8.2 g, 38 mmol, 1.8 equiv.). The reaction mixture was then cooled to 0 °C using an ice bath, followed by the addition of TEA (5.5 mL, 40 mmol, 1.9 equiv.). The reaction was stirred overnight, while slowly allowing to reach room temperature. The next morning, THF was removed *in vacuo*. The residue was dissolved in CH_2_Cl_2_ (50 mL) and then washed with H_2_O (3 × 100 mL). The aqueous layers were combined and back-extracted with CH_2_Cl_2_ (3 × 100 mL). The CH_2_Cl_2_ layers were combined and washed with brine (300 mL), dried over MgSO_4_, and then filtered. The filtrate was concentrated *in vacuo*, purified by automated flash column chromatography (100% EtOAc), and fractions evaluated by TLC and visualized by ninhydrin stain. Pure tetraethylene glycol Boc-amide **16** (3.1 g, 11 mmol, 52%) was obtained as a colourless oil. R_f_ = 0.70 (5% MeOH in CH_2_Cl_2_); ^1^H-NMR (400 MHz, CDCl_3_): δ = 5.62 (broad s, 1H, boc-N*H*), 3.69 (m, 4H, 2 × C*H*_2_), 3.62 (m, 8H, 4 × C*H*_2_), 3.52 (t, ^3^J_HH_ = 5.3 Hz, 2H, boc-NHCH_2_C*H*_2_), 3.30 (q, ^3^J_HH_ = 5.3 Hz, 2H, boc-NHC*H*_2_), 3.05 (broad s, 1H, O*H*), 1.42 (s, 9H, C(C*H*_3_)_3_) ppm; ^13^C-NMR (101 MHz, CDCl_3_): δ = 156.2 (*t*BuOO*C*), 78.9 (*C*(CH_3_)_3_), 72.6 (*C*H_2_), 70.6 (*C*H_2_), 70.5 (*C*H_2_), 70.3 (boc-NHCH_2_*C*H_2_), 70.1 (*C*H_2_), 61.7 (*C*H_2_), 40.4 (boc-NH*C*H_2_), and 28.4 (C(*C*H_3_)_3_) ppm; HRMS: calculated *m/z* for C_13_H_27_NO_6_: 316.1736 [*M*+Na]^1+^; found 316.1725.

Tetraethylene glycol Boc-amide p-toluenesulfonate (**17**). All of the steps were performed under N_2_ atmosphere. Boc-protected amino-tetraethylene glycol **16** (3.1 g, 11 mmol, 1.0 equiv.) and p-toluenesulfonyl chloride 8 (3.2 g, 17 mmol, 1.5 equiv.) were dissolved in anhydrous CH_2_Cl_2_ (50 mL). The resulting mixture was cooled to 0 °C using an ice bath, followed by the addition of TEA (2.3 mL, 17 mmol, 1.5 equiv.). The reaction was stirred overnight at room temperature. The next morning the reaction was not complete, as determined by TLC (100% EtOAc) and visualized with UV and ninhydrin stain. Nevertheless, the reaction mixture was worked-up by washing with H_2_O (3 × 50 mL). Subsequently, the combined aqueous layers were back-extracted with CH_2_Cl_2_ (2 × 100 mL). The combined CH_2_Cl_2_ layers were washed with brine (250 mL), dried over MgSO_4_, and then filtered. Next, the filtrate was concentrated *in vacuo*, purified by automated flash column chromatography using a linear gradient (50–60% EtOAc in petroleum ether 40–60 °C over 10 column volumes), and then visualized with UV and ninhydrin stain. Fractions containing pure product were combined and concentrated *in vacuo*, affording tetraethylene glycol Boc-amide p-toluenesulfonate **17** (2.1 g, 4.7 mmol, 43%) as a colourless oil. R_f_ = 0.55 (EtOAc); t_R_ = 33.0 min.; ^1^H-NMR (400 MHz, CDCl_3_): δ = 7.79 (d, ^3^J_HH_ = 8.2 Hz, 2H, aryl *o*-*H*), 7.33 (d, ^3^J_HH_ = 8.2 Hz, 2H, aryl *m*-*H*), 4.98 (broad s, 1H, boc-N*H*), 4.15 (t, ^3^J_HH_ = 4.8 Hz, 2H, TsOC*H*_2_), 3.69 (t, ^3^J_HH_ = 4.8 Hz, 2H, TsOCH_2_C*H*_2_), 3.59 (m, 8H, 4 × C*H*_2_), 3.52 (t, ^3^J_HH_ = 5.3 Hz, 2H, boc-NHCH_2_C*H*_2_), 3.29 (q, ^3^J_HH_ = 5.3 Hz, 2H, boc-NHC*H*_2_), 2.44 (s, 3H, C*H*_3_), 1.43 (s, 9H, C(C*H*_3_)_3_) ppm; ^13^C-NMR (101 MHz, CDCl_3_): δ = 156.0 (*t*BuO(O)*C*), 144.8 (aryl-*C*), 133.1 (aryl-*C*), 129.8 (aryl *m*-*C*H), 128.0 (aryl *o*-*C*H), 79.2 (*C*(*C*H_3_)_3_), 70.8 (*C*H_2_), 70.6 (*C*H_2_), 70.5 (*C*H_2_), 70.2 (*C*H_2_), 70.2 (boc-NHCH_2_*C*H_2_), 69.2 (TsO*C*H_2_), 68.7 (TsOCH_2_C*H*_2_ (7)), 40.4 (boc-NH*C*H_2_), 28.4 (C(*C*H_3_)_3_), and 21.6 (*C*H_3_) ppm; HRMS: calculated *m/z* for C_20_H_33_NO_8_S: 470.1825 [*M*+Na]^1+^; found 470.1814.

Triethylene glycol Boc-amide ethylamine (**18**). p-toluenesulfonate tetraethylene glycol with a Boc-protected amine **17** (2.1 g, 4.7 mmol, 1.0 equiv.) was dissolved in MeCN (50 mL), followed by the addition of aqueous 28–30% ammonia (50 mL). The reaction was stirred overnight at room temperature. Because the reaction was not complete, the solvents were removed *in vacuo*, after which the residue was re-dissolved in MeCN (50 mL) and fresh aqueous 28–30% ammonia (50 mL) was added. Despite repeating this procedure twice, the reaction was not complete, as determined by TLC (5% MeOH in CH_2_Cl_2_ supplemented with TEA) and visualized with ninhydrin stain. Nevertheless, solvent was removed *in vacuo* and the residue was re-dissolved in CH_2_Cl_2_ (100 mL). The resulting solution was washed with aqueous 1.0 M NaOH (3 × 100 mL) and the resulting aqueous layer was back-extracted with CH_2_Cl_2_ (3 × 100 mL). The CH_2_Cl_2_ layers were combined and washed with brine (200 mL), dried over MgSO_4_, and then filtered. Next, the filtrate was concentrated *in vacuo*, purified by automated flash column chromatography using a linear gradient (0–5% MeOH in CH_2_Cl_2_ over 10 column volumes), and visualized with UV and ninhydrin stain. Fractions containing pure product were combined and concentrated *in vacuo*, affording pure triethylene glycol Boc-amide ethylamine **18** (0.49 g, 1.7 mmol, 36%) as a colourless oil. R_f_ = 0.41 (5% MeOH in CH_2_Cl_2_ supplemented with TEA); ^1^H-NMR (400 MHz, CDCl_3_): δ = 5.26 (broad s, 1H, boc-N*H*), 3.62 (m, 8H, 4 × C*H*_2_), 3.53 (m, 4H, boc-NHCH_2_C*H*_2_ and C*H*_2_CH_2_NH_2_), 3.29 (m, 2H, boc-NHC*H*_2_), 2.88 (m, 2H, C*H*_2_NH*_2_*), 2.43 (m, 2H, N*H*_2_), 1.43 (s, 9H, C(C*H*_3_)_3_) ppm; ^13^C-NMR (101 MHz, CDCl_3_): δ = 156.1 (*t*BuO(O)*C*), 79.1 (*C*(*C*H_3_)_3_), 72.7 (boc-NHCH_2_*C*H_2_ or *C*H_2_CH_2_NH_2_), 70.5 (*C*H_2_), 70.5 (*C*H_2_), 70.2 (boc-NHCH_2_*C*H_2_ or *C*H_2_CH_2_NH_2_), 41.5 (*C*H_2_NH*_2_*), 40.4 (boc-NH*C*H_2_), and 28.4 (C(*C*H_3_)_3_) ppm; HRMS: calculated *m/z* for C_13_H_28_N_2_O_5_: 315.1896 [*M*+Na]^1+^; found 315.1883.

Triethylene glycol Boc-amide ethylamine equipped TAC-scaffold (**19**). TAC-scaffold **11** (39 mg, 50 μmol, 1.0 equiv.) was dissolved in DMF (8 mL). Next, BOP (26 mg, 59 μmol, 1.2 equiv.) was dissolved in DMF (3 × 2 mL) and then added to the solution containing TAC-scaffold **11**. Subsequently, triethylene glycol Boc-amide ethylamine 18 (29 mg, 100 μmol, 2.0 equiv.) was dissolved in DMF (3 × 2 mL) and added. Finally, D*i*PEA (35 μL, 0.20 mmol) was added and the reaction mixture was stirred for 1 h at room temperature. Upon completion of the reaction, as determined by LCMS, solvent was removed *in vacuo* (60 °C). The residue was taken up in *t*BuOH/H_2_O (1/1, *v/v*) and lyophilized. The crude yellowish oil was dissolved in CH_2_Cl_2_ and purified using automated flash column chromatography with a linear gradient (0–5% MeOH in CH_2_Cl_2_ over 10 column volumes) and then visualized with UV and ninhydrin stain. Pure fractions were combined and concentrated *in vacuo* (60 °C), the residue was taken up in *t*BuOH/H_2_O (1/1, *v/v*) and lyophilized, affording triethylene glycol Boc-amide ethylamine equipped TAC-scaffold 19 (46 mg, 43 μmol, 86%) as a white powder. R_f_ = 0.43 (5% MeOH in CH_2_Cl_2_); t_R_ = 54.2 min.; ^1^H-NMR (400 MHz, CDCl_3_): δ = 7.73 (m, 2H, aryl-*H*), 7.28 (m, 1H, aryl-*H*), 6.90 (broad s, 1H, OCH_2_CH_2_N*H*), 5.11 (broad s, 1H, boc-N*H*), 4.64 (m, 4H, 2 × NC*H*_2_-aryl), 3.65 (m, 10H, 4 × C*H*_2_ and OCH_2_C*H*_2_NH), 3.52 (t, ^3^J_HH_ = 5.2 Hz, 2H, boc-NHCH_2_C*H*_2_), 3.43 (m, 4H, NC*H*_2_CH_2_C*H*_2_N), 3.27 (q, ^3^J_HH_ = 5.2 Hz, 2H, boc-NHC*H*_2_), 2.95 (m, 6H, NC*H*_2_CH_2_C*H_2_*N and OC*H*_2_CH_2_NH), 2.64 (m, 8H, 2 × C*H*_2_C*H*_2_CCSi), 2.45 (m, 4H, C*H_2_*C*H*_2_CCH), 1.95 (m, 1H, CC*H*), 1.58 (m, 4H, 2 × NCH_2_C*H_2_*CH_2_N), 1.43 (s, 9H, C(C*H*_3_)_3_), 1.06 (m, 21H, Si(C*H*(C*H*_3_)_2_)_3_), 0.98 (t, ^3^J_HH_ = 7.9 Hz, 9H, Si(CH_2_C*H*_3_)_3_), 0.58 (q, ^3^J_HH_ = 7.9 Hz, 6H, Si(C*H*_2_CH_3_)_3_) ppm; ^13^C-NMR (101 MHz, CDCl_3_): δ = 156.0 (*t*BuO(O)*C*)), 129.2 (aryl-*C*H), 126.8 (aryl-*C*H), 70.5 (*C*H_2_), 70.5 (*C*H_2_), 70.3 (*C*H_2_), 70.2 (boc-NHCH_2_*C*H_2_ or O*C*H_2_CH_2_NH), 70.2 (boc-NHCH_2_*C*H_2_ or O*C*H_2_CH_2_NH), 70.1 (*C*H_2_), 53.9 (N*C*H_2_-aryl), 51.9 (NC*H*_2_-aryl), 48.0 (N*C*H_2_CH_2_*C*H_2_N or O*C*H_2_CH_2_NH), 45.9 (N*C*H_2_CH_2_*C*H_2_N or OCH_2_*C*H_2_NH), 45.7 (N*C*H_2_CH_2_*C*H_2_N or OCH_2_*C*H_2_NH), 45.3 (N*C*H_2_CH_2_*C*H_2_N or O*C*H_2_CH_2_NH), 43.7 (N*C*H_2_CH_2_*C*H_2_N or O*C*H_2_CH_2_NH), 40.3 (boc-NH*C*H_2_), 39.8 (*C*H_2_), 33.0 (*C*H_2_CH_2_CCSi), 33.0 (*C*H_2_CH_2_CCSi), 32.0 (*C*H*_2_*CH_2_CCH), 28.4 (C(*C*H_3_)_3_), 28.0 (NCH_2_*C*H_2_CH_2_N), 28.0 (NCH_2_*C*H_2_CH*_2_*N), 18.6 (Si(CH(*C*H_3_)_2_)_3_), 16.3 (*C*H_2_*C*H_2_CCSi), 16.3 (*C*H_2_*C*H_2_CCSi), 14.5 (*C*H*_2_C*H_2_CCH), 11.3 (Si(*C*H(CH_3_)_2_)_3_), 7.5 (Si(CH_2_*C*H_3_)_3_), 4.5 (Si(*C*H_2_CH_3_)_3_) ppm; HRMS: calculated *m/z* for C_58_H_95_N_5_O_9_Si_2_: 1084.6566 [*M*+Na]^1+^; found 1084.6559; LRMS: calculated *m/z* for C_58_H_95_N_5_O_9_Si_2_:1062.67 [*M*+H]^1+^/1084.66 [M+Na]^1+^; Found *m/z* 1062.83/1084.75.

### 2.4. A General Method Employing CuAAC towards Continuous Epitope Mimics 20, 21, and 22

Continuous epitope mimics **20** and **21** were synthesized, as described earlier by Meuleman et al. [33].

Continuous epitope mimic **22**. Cyclic peptide **6** (29 mg, 12 μmol) and linker^33^ (16 mg, 33 μmol; dissolved in 190 μL DMF of which 100 μL was used) was subjected to the above described CuAAC with CuSO_4_•5H_2_O (8.3 mg, 33 μmol; dissolved in 160 μL H_2_O of which 100 μL was used), sodium L-ascorbate (28 mg, 0.14 mmol; dissolved in 122 μL H_2_O of which 100 μL was used), tris[(1-benzyl-1H-1,2,3-triazol-4-yl)methylamine (TBTA) (5.2 mg, 9.8 μmol; dissolved in 170 μL DMF of which 100 μL was used), aminoguanidine•HCl (17 mg, 0.15 mmol; dissolved in 122 μL of which 100 μL was used). Purification and subsequent deprotection afforded continuous epitope mimic **22** (17 mg, 6.2 μmol, 52%). t_R_ = 17.8 min.; HRMS: calculated *m/z* for C_116_H_170_N_34_O_38_S_3_: 1372.5867 ½[*M*+2H]^2+^; found 1372.5808; LRMS: calculated *m/z* for C_116_H_170_N_34_O_38_S_3_: 1372.59 ½[*M*+2H]^2+^/915.39 1/3[*M*+3H]^3+^; found 1373.75/916.58.

### 2.5. Assembly of the Discontinuous Epitope Mimic **27**


The cycloaddition of the first cyclic peptide and subsequent triethylsilyl (TES) deprotection. Triethylene glycol Boc-amide ethylamine equipped TAC-scaffold **19** (66 mg, 62 μmol, 1.8 equiv.) was dissolved in DMF (600 μL) and added to cyclic peptide 5 (71 mg, 34 μmol, 1.0 equiv.). The flask-containing TAC-scaffold **19** was rinsed with additional DMF (500 μL), which was added to the reaction mixture to ensure the complete transfer of **19**. Next, TBTA (15 mg, 28 μmol) was dissolved in DMF (161 μL), of which 100 μL (17 μmol, 0.50 equiv.) was added to the reaction mixture. Subsequently, aqueous solutions of CuSO_4_•5H_2_O (24 mg, 97 μmol, dissolved in 158 μL H_2_O), sodium L-ascorbate (68 mg, 0.34 mmol, dissolved in 202 μL H_2_O), and aminoguanidine•HCl (59 mg, 0.53 mmol, dissolved in 142 μL H_2_O) were prepared. The resulting aqueous solutions were mixed (CuSO_4_•5H_2_O (100 μL, 61 μmol, 1.8 equiv.), sodium L-ascorbate (200 μL, 0.34 mmol, 10 equiv.), and aminoguanidine•HCl (100 μL, 0.38 mmol, 11 equiv.)), and the resulting mix (400 μL) was added dropwise to the stirring reaction mixture. The reaction was kept under N_2_-atmosphere and monitored by LCMS. After 3 h, the reaction was complete, followed by the addition of Cuprisorb resin to capture the Cu-ions. The reaction mixture was filtered and the reaction vessel and filter were washed with extra DMF (5 × 2 mL). Next, DMF was removed *in vacuo* (60 °C) and the residue was dissolved in *t*BuOH/H_2_O/MeCN (1/1/1, *v/v/v*), and lyophilized.

The resulting crude product was dissolved in DMF (1 mL) and an aqueous solution of AgNO_3_ (0.25 g, 1.5 mmol, dissolved in 235 μL H_2_O) was prepared, of which 100 μL (0.62 mmol, 18 equiv.) was added. The reaction was stirred under N_2_-atmosphere for 1 h at room temperature and monitored by LCMS. After 1 h, the reaction was not complete and an additional 100 μL (0.62 mmol, 18 equiv.) was added. The reaction was complete after an additional hour of stirring, as was verified by LCMS. Next, additional DMF (900 μL) was added to the reaction mixture and an aqueous solution of NaCl (0.12 g, 2.1 mmol, dissolved in 510 μL H_2_O) was prepared, of which 300 μL (1.2 mmol, 35 equiv.) was added to the reaction mixture. After 15 min., the reaction mixture was collected, and the reaction vessel rinsed with DMF (3 × 2 mL). The resulting AgCl precipitate was removed by centrifugation (4500 rpm; 5 min.), the resulting precipitate was washed with DMF (5 mL), and it was centrifuged again. The supernatants were collected and pooled, followed by removal of DMF *in vacuo* (60 °C). The residue was dissolved in *t*BuOH/H_2_O/MeCN (1:1:1, *v/v/v*) and lyophilized. 

The obtained crude product was dissolved in buffer 4 mL A/B (1/1, *v/v*), clarified by centrifugation (4500 rpm; 5 min.), and purified by preparative HPLC while using a customary protocol: 50% buffer B in buffer A for 5 min., followed by a linear gradient of 50–100% buffer B in buffer A for 50 min. Fractions containing pure product were identified by analytical HPLC, pooled, and lyophilized, which afforded pure scaffold molecular construct **23** (27 mg, 8.9 μmol, 26%). t_R_ = 27.5 min.; HRMS: calculated *m/z* for C_150_H_208_N_32_O_31_S_2_Si: 1523.7525 ½[*M*+2H]^2+^; found 1523.7543; LRMS: calculated *m/z* for C_150_H_208_N_32_O_31_S_2_Si: 1523.75 ½[*M*+2H]^2+^/1016.17 1/3[*M*+3H]^3+^; found 1524.83/1016.92.

The cycloaddition of the second cyclic peptide. Cyclic peptide **6** (50 mg, 20 μmol, 1.5 equiv.) was dissolved in DMF (500 μL) and added to scaffold molecular construct 23 (40 mg, 13 μmol, 1.0 equiv.). The flask-containing cyclic peptide 6 was rinsed with additional DMF (500 μL), which was added to the reaction mixture to ensure complete transfer. Next, TBTA (9.3 mg, 18 μmol) was dissolved in DMF (265 μL), of which 100 μL (6.6 μmol, 0.51 equiv.) was added to the reaction mixture. Subsequently, aqueous solutions of CuSO_4_•5H_2_O (12 mg, 48 μmol, dissolved in 198 μL H_2_O), sodium L-ascorbate (52 mg, 0.26 mmol, dissolved in 200 μL H_2_O), and aminoguanidine•HCl (32 mg, 0.29 mmol, dissolved in 200 μL H_2_O) were prepared. The resulting aqueous solutions were mixed (CuSO_4_•5H_2_O (100 μL, 24 μmol, 1.8 equiv.), sodium L-ascorbate (100 μL, 0.13 mmol, 10.0 equiv.), and aminoguanidine•HCl (100 μL, 0.15 mmol, 11 equiv.)) and the resulting mix (300 μL) was added dropwise to the stirring reaction mixture. The reaction was kept under N_2_-atmosphere and monitored by LCMS. After 6 h, the reaction was complete; this was followed by the addition of Cuprisorb resin to capture the Cu-ions. The reaction mixture was filtered. The reaction vessel and filter were washed with extra DMF (5 × 2 mL). Next, DMF was removed *in vacuo* (60 °C), the residue was dissolved in *t*BuOH/H_2_O/MeCN (1/1/1, *v/v/v*), and then lyophilized.

The obtained crude product was dissolved in buffer 5 mL A/B (1/1, *v/v*), clarified by centrifugation (4500 rpm; 5 min.), and the resulting compound in the supernatant was purified by preparative HPLC using a customary protocol: 40% buffer B in buffer A for 5 min., followed by a linear gradient of 40–70% buffer B in buffer A for 60 min. Fractions containing pure product were identified by analytical HPLC, pooled, and lyophilized, which afforded pure scaffold molecular construct 24 (17 mg, 3.0 μmol, 23%). t_R_ = 23.7 min.; HRMS: calculated *m/z* for C_255_H_358_N_66_O_65_S_4_Si: 1386.1425 1/4[*M*+4H]^4+^; found 1386.1481; LRMS: calculated *m/z* for C_255_H_358_N_66_O_65_S_4_Si: 1847.85 1/3[*M*+3H]^3+^/1386.14 1/4[*M*+4H]^4+^; found 1849.00/1387.17.

Triisopropylsilyl (TIPS) deprotection. Tetra-n-butylammonium fluoride (TBAF)•3H_2_O (13 mg, 41 μmol) was dissolved in DMF (133 μL), of which 100 μL (31 μmol, 10 equiv.) was used to dissolve the scaffold molecular construct **24** (17 mg, 3.0 μmol, 1.0 equiv.). The reaction was stirred under N_2_-atmosphere and closely monitored by LCMS. After 5 h, no conversion was observed and fresh TBAF•3H_2_O (14 mg, 44 μmol; dissolved in 152 μL DMF, of which 100 μL (29 μmol, 10 equiv.)) was added. The next day, the reaction progressed, but it was still not complete. Therefore, fresh TBAF•3H_2_O (24 mg, 76 μmol; dissolved in 255 μL DMF, of which 100 μL (29.8 μmol, 9.9 equiv.)) was added. The next morning, the reaction was complete, as verified by LCMS. The reaction mixture was collected and the reaction vessel was rinsed with 5 mL buffer A:B (1:1, *v/v*). Next, the reaction mixture was clarified by centrifugation (4500 rpm; 5 min.) and then purified by preparative HPLC using a customary protocol: 20% buffer B in buffer A for 5 min., followed by a linear gradient of 20–50% buffer B in buffer A for 40 min. Fractions containing pure product were identified by analytical HPLC, pooled, and lyophilized, which afforded pure scaffolded molecular construct **25** (6.6 mg, 1.2 μmol, 40%). t_R_ = 20.4 min.; HRMS: calculated *m/z* for C_246_H_338_N_66_O_65_S_4_: 1347.1092 1/4[*M*+4H]^4+^; found 1347.1139; LRMS: calculated *m/z* for C_246_H_338_N_66_O_65_S_4_: 1795.81 1/3[*M*+3H]^3+^/1347.11 1/4[*M*+4H]^4+^; found 1797.17/1348.42.

The cycloaddition of the third cyclic peptide. Cyclic peptide **4** (4.4 mg, 2.1 μmol, 1.8 equiv.) was dissolved in DMF (500 μL) and added to scaffold molecular construct **25** (6.6 mg, 1.2 μmol, 1.0 equiv.). The flask-containing cyclic peptide **4** was rinsed with additional DMF (500 μL), which was added to the reaction mixture to ensure complete transfer. Next, TBTA (1.1 mg, 2.0 μmol) was dissolved in DMF (330 μL), of which 100 μL (0.60 μmol, 0.50 equiv.) was added to the reaction mixture. Subsequently, aqueous solutions of CuSO_4_•5H_2_O (3.0 mg, 12 μmol, dissolved in 555 μL H_2_O), sodium L-ascorbate (15 mg, 76 μmol, dissolved in 635 μL H_2_O), and aminoguanidine•HCl (14 mg, 0.13 mmol; dissolved in 950 μL H_2_O) were prepared. The resulting aqueous solutions were mixed ((CuSO_4_•5H_2_O (100 μL, 2.2 μmol, 1.8 equiv.), sodium L-ascorbate (100 μL, 12 μmol, 10 equiv.), and aminoguanidine•HCl (100 μL, 14 μmol, 12 equiv.)), and the resulting mix (300 μL) was added dropwise to the stirring reaction mixture. The reaction was kept under N_2_-atmosphere and monitored by LCMS. After 3 h, no more cyclic peptide **4** was observed in the reaction mixture but scaffold molecular construct **25** was still present. Therefore, additional cyclic peptide **4** (2.7 mg, 1.3 μmol, 1.1 equiv.) was dissolved in DMF (100 μL) and added. After 1 h, no improvement was observed by LCMS and fresh reagents were prepared. TBTA (1.1 mg, 2.0 μmol) was dissolved in DMF (330 μL), of which 100 μL (0.6 μmol, 0.5 equiv.) was added to the reaction mixture. Subsequently, solutions of CuSO_4_•5H_2_O (6.0 mg, 24 μmol; dissolved in 555 μL H_2_O), sodium L-ascorbate (7.9 mg, 40 μmol; dissolved in 166 μL H_2_O), and aminoguanidine•HCl (11 mg, 99 μmol, dissolved in 370 μL H_2_O) were prepared. The resulting solutions were mixed ((CuSO_4_•5H_2_O (50 μL, 2.2 μmol, 1.8 equiv.), sodium L-ascorbate (50 μL, 12 μmol, 10 equiv.), and aminoguanidine•HCl (100 μL, 13 μmol, 11.0 equiv.)) and the resulting mix (150 μL) was added dropwise to the stirring reaction mixture. The reaction was continued to stir under N_2_-atmosphere and monitored by LCMS. After 2 h, additional product was formed, but the reaction was not fully complete. Nevertheless, the reaction was worked-up by the addition of Cuprisorb resin to capture the Cu-species. The reaction mixture was filtered, and the reaction vessel and filter were washed with extra DMF (5 × 2 mL). Next, DMF was removed *in vacuo* (60 °C), the residue was dissolved in *t*BuOH/H_2_O/MeCN (1/1/1, *v/v/v*), and lyophilized.

The obtained crude product was suspended in buffer 3 mL A:B (1:1, *v/v*), clarified by centrifugation (4500 rpm; 5 min.), and the resulting compound in the supernatant was purified by preparative HPLC using a customary protocol: 0% buffer B in buffer A for 5 min., followed by a linear gradient of 0–50% buffer B in buffer A for 50 min. Fractions containing pure product were identified by analytical HPLC, pooled, and lyophilized, which afforded pure discontinuous epitope mimic **26** (2.2 mg, 0.29 μmol, 24%). t_R_ = 19.6 min.; HRMS: calculated *m/z* for C_334_H_475_N_99_O_90_S_6_: 1501.8870 1/5[*M*+5H]^5+^; found 1501.8881; LRMS: calculated *m/z* for C_334_H_475_N_99_O_90_S_6_: 1877.11 1/4[*M*+4H]^4+^/1501.89 1/5[*M*+5H]^5+^/1251.74 1/6[*M*+6H]^6+^; found 1878.42/1503.00/1252.58.

The removal of Boc protective group of **26**. Discontinuous epitope mimic **26** (2.2 mg, 0.29 μmol, 1.0 equiv.) was dissolved in 250 μL TFA/H_2_O/TIS/EDT (90/5/2.5/2.5, *v/v/v/v*) and the reaction mixture was stirred for 30 min. at room temperature. Next, the reaction mixture was added to Et_2_O (15 mL) to precipitate the product. The reaction vessel was rinsed (x3) with an additional 250 μL TFA/H_2_O/TIS/EDT (90/5/2.5/2.5, *v/v/v/v*), which was added to a separate volume of Et_2_O (15 mL) to collect all of the product. Precipitated product was obtained by centrifugation (4500 rpm; 5 min.), and washed twice using Et_2_O (15 mL), followed by centrifugation (4500 rpm; 5 min.). The obtained precipitated pellets were dissolved in *t*BuOH/H_2_O/MeCN (1/1/1, *v/v/v*), pooled, and lyophilized, which afforded pure discontinuous epitope mimic **27** (2.0 mg, 0.27 μmol, *quant.*). t_R_ = 18.9 min.; HRMS: calculated *m/z* for C_329_H_467_N_99_O_88_S_6_: 1481.8765 1/5[*M*+5H]^5+^; found 1481.8699; LRMS: calculated *m/z* for C_329_H_467_N_99_O_88_S_6_: 1852.09 1/4[*M*+4H]^4+^/1481.88 1/5[*M*+5H]^5+^/1235.07 1/6[*M*+6H]^6+^; found 1853.42/1483.00/1236.17.

Conjugation of free thiol containing continuous epitope mimics (**20**, **21**, and **22**) on maleimide activated macriculture keyhole limpet hemocyanin (mcKLH). This procedure was performed, as instructed by the Imject^®^ Maleimide Activated Carrier Protein Spin Kits purchased from Thermo Scientific.

One vial of the activated mcKLH was reconstituted per conjugation by adding 200 μL ultrapure H_2_O to obtain a 10 mg/mL translucent to whitish-blue solution. Next, 2.0 mg of continuous epitope mimic (**20**, **21**, and **22**) were separately dissolved in 500 μL conjugation buffer (83 mM sodium phosphate, 0.10 M EDTA, 0.90 M sodium chloride, 0.10 M sorbitol, and 0.02% sodium azide; pH 7.2). Immediately, each solution of continuous epitope mimic was mixed with one vial of activated mcKLH were mixed and incubated for 2 h at room temperature.

After conjugation, 10 mL of ultrapure H_2_O was added to one bottle of Imject purification buffer salts (83 mM sodium phosphate, 0.90 M sodium chloride, 0.10 M sorbitol; pH 7.2) per conjugation sample. One desalting column was drained by centrifugation (1000× *g*; 2 min.) per conjugation sample. The desalting columns were prepared by slowly adding 1 mL of purification buffer and draining by centrifugation (1000× *g*; 2 min.) for a total of four times. Each conjugation sample was collected and centrifuged (1000× *g*; 2 min.), the pellets were kept, and the resulting supernatants (700 μL) were carefully loaded on separate desalting columns. Each desalting column was placed in a clean collection tube and centrifuged (1000× *g*; 2 min.) to collect the conjugation samples. The collected conjugation samples were used to resuspend their corresponding pellets. The obtained continuous epitope-mcKLH conjugates (**20**-mcKLH, **21**-mcKLH, and **22**-mcKLH) were stored at −20 °C until immunization.

Conjugation of free amine containing discontinuous epitope mimic (**27**) on mcKLH by 1-ethyl-3-[3-dimethylaminopropyl]carbodiimide hydrochloride (EDC) coupling. This procedure was performed, as instructed by the Imject^®^ EDC Carrier Protein Spin Kits tha were purchased from Thermo Scientific.

One vial of mcKLH was reconstituted by adding 200 μL ultrapure H_2_O to obtain a 10 mg/mL translucent to a whitish-blue solution. Next, 2.0 mg of discontinuous epitope mimic **27** was dissolved in 450 μL Imject^®^ EDC conjugation buffer (0.10 M MES, 0.90 M sodium chloride, 0.02% sodium azide; pH 4.7) and it was added to the mcKLH solution. After which, one vial of EDC (10 mg) was dissolved in 1 mL ultrapure H_2_O, of which 50 μL was immediately added to the epitope-mcKLH mixture. Subsequently, the mixture was incubated for 2 h at room temperature.

After the conjugation, 10 mL of ultrapure H_2_O was added to one bottle of Imject purification buffer salts (83 mM sodium phosphate, 0.90 M sodium chloride, 0.10 M sorbitol; pH 7.2). One desalting column was drained by centrifugation (1000× *g*; 2 min.) and was prepared by slowly adding 1 mL of purification buffer and draining by centrifugation (1000× *g*; 2 min.) for a total of four times. The conjugation sample was collected and centrifuged (1000× *g*; 2 min.), the pellet was kept, and the resulting supernatant (650 μL) was carefully loaded on the desalting column. The desalting column was placed in a clean collection tube and centrifuged (1000× *g*; 2 min.) to collect the conjugation sample. The collected conjugation sample was used to resuspend the obtained pellet and the resulting discontinuous epitope-mcKLH conjugate (**27**-mcKLH) was stored at −20 °C until immunization.

*Immunization experiments*. The mouse immunization experiments were approved by the University of Glasgow Animal Welfare and Ethical Board and they were carried out under the Home Office Project License P9722FD8E held by Prof. Arvind H. Patel at the MRC-University of Glasgow Centre for Virus Research.

Two independent immunization experiments were performed. Each experiment had four groups that contained three mice per group that were subcutaneously injected with epitope-mcKLH conjugate (**20**-mcKLH, **21**-mcKLH, **22**-mcKLH, or **27**-mcKLH) mixed with Addavax (Invivogen) adjuvant (350 μL). Prime immunization was performed with 50 μg (131 μL) epitope-mcKLH conjugate (20-mcKLH, 21-mcKLH, 22-mcKLH, or 27-mcKLH). After two weeks, the prime immunization was followed up by one boost immunization of 10 μg (26 μL) epitope-mcKLH conjugate (**20**-mcKLH, **21**-mcKLH, **22**-mcKLH, or **27**-mcKLH) every week for four weeks. The animals were sacrificed eight days after the last boost immunization and sera were collected for binding and neutralization assays.

*Monoclonal antibodies*. The mAb AP33 [21] and 1:7 [35] used in this work were previously described.

*General method for ELISA*. *Plate preparation:* Immulon 2HB 96-well plates were coated with 1 μg/mL sE2 Gt1a H77 (purified from mammalian HEK-F cells) overnight at room temperature. After which, the wells were blocked overnight at room temperature with 200 μL 2% skimmed milk in PBST (phosphate buffered saline supplemented with 0.05% Tween^®^-20). Next, the wells were washed three times with PBST and then used immediately or stored at −20 °C until use.

*Sera preparation:* the obtained mice blood from the immunization experiment was incubated for 1 h at 37 °C and the tubes were flicked to dislodge the blood clot. Subsequently, the tubes were incubated at 4 °C for 2 h or overnight to enhance blood clotting. After which, the blood clots were centrifuged (10,000× *g*; 10 min) at 4 °C. Sera were collected from the blot clots and then transferred to a fresh tube. The centrifugation/transfer step was repeated if necessary.

*ELISA:* three-fold dilution series of sera were prepared starting from a 1:50 dilution (3 μL sera and 147 μL PBST) and then added to the plates (100 μL/well). Monoclonal antibody AP33 was included as a positive control (three-fold dilution at starting concentrations: 2.0 μg/mL and 0.1 μg/mL). The plates were incubated for 1–2 h at room temperature. After which, the plates were washed three times with PBST, before supplying 100 μL 1:3000 secondary α-mouse A4416 (Sigma) antibodies to each well. The plates were incubated for 1 h at room temperature and washed six times with PBST. The plates were developed using 100 μL 3, 3′, 5, 5′ tetramethylbenzidine (TMB) solution per well, obtained from Life Technologies, and incubated for 10 min. at room temperature. After which, development was stopped using 50 μL 0.5 M H_2_SO_4_ per well. Absorbance at 450 nm was measured on a Varioskan (Thermoscientific) instrument. 

*Cell lines*. human hepatoma Huh-7 cells and HEK-293T cells were grown in Dulbecco’s modified Eagle’s medium that was supplemented with 10% fetal calf serum, 5% nonessential amino acids, Penicillin, Streptomycin, and 200 mM L-glutamine.

*Generation of HCV pseudoparticles (HCVpp).* HEK-293T cells were co-transfected with plasmids expressing MLV Gag-pol, the MLV transfer vector carrying firefly luciferase reporter and HCV E1E2 glycoprotein. After 72 h, the medium was harvested, filtered through a 0.45 µm membrane, and used as a source of HCVpp, as previously described [36,37].

*Neutralization assays*. IgG was purified from sera using the NAb™ Protein A/G spin columns (Thermofisher). Huh-7 cells were seeded at 4 × 10^3^ cells per well in a 96-well plate. Three-fold dilution series of purified IgG were prepared starting from 100 μg/mL. Monoclonal antibodies AP33 and 1:7 were included as positive controls (three-fold dilution series starting from 50 μg/mL). 45 μL of Gt1a H77 HCVpp were added to each well and then incubated for 1 h at 37 °C. Next, Huh-7 cells were inoculated for 3 h at 37 °C. After which, the inoculum was removed and 120 μL of fresh media was added. The cells were lysed after an incubation period of 72 h at 37 °C post-inoculation. Infectivity and neutralization were assessed using the GloLysis Luciferase assay (Promega).

## 3. Results and Discussion

### 3.1. Synthesis and Cyclization of Cysteine Containing Peptides Based on HCV-E2 Glycoprotein

A highly conserved discontinuous epitope is located within the cellular receptor CD81 binding site of HCV-E2 glycoprotein that consists of four peptide segments [26]. Individually, these peptide segments are referred to as epitopes I, II, III, and IV. So far, we have shown the antigenic potential of monocyclic peptides based on epitopes I, II, and III, which were conjugated onto a maleimide-activated surface, using ELISA plates [32,33]. Epitope IV has not been considered to be a continuous epitope, as it combines with epitopes I, II, and III to form a discontinuous epitope [26]. We decided to omit epitope IV because only epitopes I, II, and III were previously included as epitope mimics. Epitope I includes residues 411–424 and it is considered a highly conserved, but flexible, immunogenic region [21,22,23]. Epitope II comprises a highly conserved immunogenic region that includes residues 436–448 [24,25]. Epitope III consists of a promising immunogenic region between residues 521–541 [26,27]. For epitope III, we previously focused on a peptide segment spanning residues 528–541 that was successfully recognized by non-neutralizing antibody DAO5 [26,33]. However, recent work conducted by Cowton *et al.* [26] showed key binding amino acid residues within epitope III that were involved in antibody binding of the combined I-II-III discontinuous epitope. Therefore, the peptide sequence of epitope III was changed to residues 521–537. Peptides **1**, **2**, and **3**, corresponding to epitopes I, II, and III, respectively, were synthesized by SPPS [17,18], with flanking N- and C-terminal cysteine residues (Table 1). These cysteine residues enabled the cyclization of the synthesized peptides onto our polar hinge [34], leading to cyclic peptides **4**, **5**, and **6,** as was described earlier by us [33] to resemble the loop-like structure of the epitope I, II, and III, respectively, as are found in the HCV E2 glycoprotein (Figure 1).

### 3.2. Adapting the Triazacyclophane Scaffold for Subsequent Conjugation

The synthesis of monocyclic peptide constructs with a flexible linker and a free thiol-moiety that allowed for thiol-maleimide conjugation provided inspiration for a similar approach that was applied to the TAC scaffold (Scheme 1) [31,32,33].

Previously developed tetraethylene glycol monotrityl thioether **7** [32] was reacted with p-toluenesulfonyl chloride **8** to obtain tosylate **9** in good yield (74%). Next, the tosylate 9 was reacted to amine 10 by treatment with aqueous ammonia. Despite that aqueous ammonia was not very effective in substituting the tosylate, as was evident from the low yield of amine 10 (14%), it does have a practical advantage in providing a simple one-step reaction. In contrast to exchanging the tosylate by an azide, followed by the sequential reduction of the azide to form an amine. TAC-scaffold **11** was synthesized according to Werkhoven et al. [20,31]. Triethylene glycol monotrityl thioether ethylamine **10** was coupled to TAC-scaffold 11 via amide bond formation using BOP, affording an orthogonally protected trialkyne TAC-scaffold 12 (81%) that, upon liberation of the thiol-moiety, could be further conjugated to a maleimide-activated mcKLH carrier protein.

Surprisingly, when functionalized to the TAC-scaffold, the trityl-protection group did not appear to be compatible with CuAAC. Therefore, it was decided to proceed to functionalize the TAC scaffold with an orthogonal protected Boc-group containing amide linker that, upon deprotection, could be conjugated to carboxyl groups in a carrier protein (Scheme 2), for example, in an EDC-coupling reaction.

Thus, mono-tosylated triethylene glycol (TEG) **13** was synthesized, as previously described by Meuleman et al. [32] Next, mono-tosylated TEG **13** was converted to a free amine **14** with aqueous ammonia. Amine **14** was then immediately protected with a Boc-group using di-*tert*-butyl dicarbonate **15** that afforded TEG Boc-amide **16** (52%) in a one-pot reaction. TEG Boc-amide **16** was then further reacted with p-toluenesulfonyl chloride **8** to obtain tosylate **17** (43%). Subsequently, tosylate **17** was converted to a free amine using aqueous ammonia to obtain triethylene glycol Boc-amide ethylamine linker **18** (36%). Next, TAC-scaffold **11** was equipped with triethylene glycol Boc-amide ethylamine linker **18** via amide bond formation while using BOP coupling that afforded orthogonally protected trialkyne TAC-scaffold **19** (86%) and after the subsequent CuAAC click reactions and deprotection reactions, it was planned to couple the resulting synthetic vaccine molecular construct to a carrier protein.

### 3.3. Assembly of the Continuous and Discontinuous Epitope Mimics by CuAAC of the Cyclic Peptides 

Continuous epitope mimics **20**, **21**, and **22** were synthesized by the conjugation of cyclic peptide **4**, **5**, or **6**, respectively, as was described by Meuleman et al. [33] (Figure 2). Upon liberating the reactive thiol-group, continuous epitope mimics **20**, **21**, and **22** could be conjugated to maleimide-activated mcKLH carrier protein.

The mimicry of discontinuous epitopes was, of course, more complicated and required the selective incorporation of all three cyclic peptides **4**–**6** onto TAC-scaffold **19** while using CuAAC. The orthogonal process of equipping the TAC-scaffold with three cyclic peptides was described by Werkhoven et al. [20,31] and it was adapted for this research (Scheme 3). The final deprotection resulted in a free amine-group that could be used to conjugate discontinuous epitope mimic **27** to mcKLH via amide bond formation. Thus, cyclic peptide 5 was coupled to the first free alkyne, followed by the removal of the TES protection group by adding silver nitrate that afforded molecular construct **23** (26%, after purification). The incorporation of the second cyclic peptide 6 onto the deprotected alkyne afforded scaffold molecular construct **24** (23%, after purification). Next, the TIPS protection group was removed using tetrabutylammonium fluoride trihydrate (TBAF•3H_2_O) to afford scaffold molecular construct **25** (40%, after purification). The resulting deprotected alkyne was used to incorporate cyclic peptide 4, which afforded discontinuous epitope mimic **26** (24%, after purification). Treatment with trifluoro acetic acid (TFA) removed the Boc-protection group and afforded discontinuous epitope mimic **27** (quantitatively, after precipitation) with no additional purification being required.

### 3.4. Eliciting Anti-HCV-E2 Glycoprotein Antibodies Using Continuous and Discontinuous Epitope Mimics

The biological potential of our continuous **20**–**22** and discontinuous **27** epitope mimics as synthetic vaccines was investigated in immunization experiments. The first question to address was if the synthetic vaccine constructs were capable of eliciting an anti-HCV E2 glycoprotein antibody response in vivo. To investigate this, the epitope mimics had to be conjugated first to a suitable carrier protein, via the incorporated thiol- and amine- linkers on the epitope mimics. The stable and efficient carrier protein mcKLH, which is known to produce a strong immunogenic response, was used for this purpose [39].

Continuous epitope mimics **20**, **21**, and **22** were conjugated to maleimide-activated mcKLH protein through thiol-maleimide conjugation using an Imject^®^ maleimide activated carrier protein spin kit (Figure 3). As described above, discontinuous epitope mimic **27** could not be obtained with a similar thiol linker, and it was prepared with a free amine instead, which was conjugated on mcKLH via amide bond formation using an Imject^®^ 1-ethyl-3-[3-dimethylaminopropyl]carbodiimide hydrochloride (EDC) carrier protein spin kit (Figure 4). Thiol-maleimide conjugation does not result in cross-linking of the constructs and carrier-protein; instead, each maleimide reacts selectively and exclusively with one copy of epitope mimic. In contrast, in conjugation via amide bond formation, considerable cross-linking within the carrier-protein and discontinuous epitope mimic **27** is possible through numerous available carboxylic acids and free-amines on mcKLH. In addition to the installed amine containing linker, discontinuous epitope mimic **27** contains a free N-terminus in cyclic peptide **4** and a Lysine residue at position 446 in cyclic peptide **5**, which are, also in principle, susceptible to amide bond formation by EDC. This may prevent the selective conjugation of discontinuous epitope mimic **27** to mcKLH occurring only via the amine containing linker. Nevertheless, it is not unlikely that conjugation to mcKLH predominantly occurs to the most available exposed amine, which is the amine containing linker.

Sera were obtained from two independent immunization experiments of four groups of animals (*n* = number of animals per group) treated with epitope mimics **20**-mcKLH (group O, *n* =3; group W, *n* = 3), **21**-mcKLH (group P, *n* = 3; group X, *n* = 3), **22**-mcKLH (group Q, *n* = 3; group Y, *n* = 3), and **27**-mcKLH (group R, *n* = 3; group Z, *n* = 3). However, one animal that was a member of group R expired due to causes that are unrelated to the immunization. Sera were evaluated by ELISA against surfaces coated with HCV soluble E2 (sE2) glycoprotein for the presence of anti-HCV E2 glycoprotein antibodies (Figure 5). Broadly, neutralizing monoclonal antibody AP33 [21] was included within the ELISA as a positive control. 

The signal that was observed for binding of monoclonal antibody (mAb) AP33 as a positive control has to be considered carefully when comparing to the signal observed for binding of antibodies in the obtained sera. Because mAb AP33 is available in a stock solution of known concentration, it was possible to generate a distinct serial dilution of set concentration ranges. In contrast, the anti-E2 antibody concentration in the obtained sera was unknown and, therefore, not directly comparable to mAb AP33.

Epitope mimic **20**-mcKLH successfully elicited antibodies against the HCV E2 glycoprotein in both group O (Figure 5A) and group W (Figure 5B). However, the response of only one animal W1 was somewhat comparable to broadly neutralizing mAb AP33. In contrast, epitope mimic **21**-mcKLH did not stimulate an anti-HCV E2 glycoprotein response in either group P (Figure 5A) or group X (Figure 5B). A strong anti-HCV E2 antibody response was found for epitope mimic **22**-mcKLH in all six animals (Figure 5), all of which were comparable to the mAb AP33 positive control. Unfortunately, immunization with discontinuous epitope mimic **27**-mcKLH did not elicit antibodies against the HCV-E2 glycoprotein (Figure 5).

The second question to address was whether purified IgG obtained from the sera could prevent HCV-infection and, thus, were capable of neutralization of HCV (Figure 6). Neutralization experiments using HCV pseudo-particles of strain H77 (HCVpp) [36,37] were performed in order to determine the level of neutralization of the obtained antibodies. Because sera obtained from immunization with **20**-mcKLH contained anti-HCV-E2 glycoprotein antibodies (Figure 5), mAb AP33 [21] was included as a positive control, as it is known to bind epitope I. Likewise, immunization with **22**-mcKLH, incorporating epitope III, also elicited antibodies capable of recognizing HCV E2 glycoprotein (Figure 5). Therefore, mAb 1:7 [26,35], known to bind epitope III, was included as a positive control. Despite the fact that immunization with epitope mimics **21**-mcKLH and **27**-mcKLH did not stimulate antibodies against HCV-E2 glycoprotein, the sera were included in the neutralization experiment. Perhaps the antibody concentration in the sera was simply too low to show binding, but perhaps neutralization of the virus still could take place. However, none of the obtained sera were capable of neutralization HCVpp (Figure 6).

The area of epitope mimicry is very challenging, and many aspects have to be considered with respect to epitope design. The optimization of the synthetic peptide sequence, length, loop size, and location of the non-native cysteines for cyclization may lead to improved mimicry. Epitope mimic **22**-mcKLH did result in a strong in vivo anti-HCV E2 antibody response. Its peptide sequence corresponded to an elongated version based on epitope III that was shifted more towards the N-terminus of the HCV-E2 glycoprotein and included more key-binding residues that are associated with neutralizing antibodies [26]. The epitope design of **20** and **21** presented here was rationalized based on previous in vitro studies [32,33], which showed that accurate mimicry was achieved, with the non-native cysteine residues being incorporated at the N-terminus and C-terminus in order to preserve the native continuous amino acid sequence. Therefore, the question remains as to why these epitope designs did not translate to an equally strong elicitation of an in vivo anti-HCV E2 antibody response for **20**-mcKLH and no elicitation of an antibody response for **21**-mcKLH.

One aspect that might play a role is that HCV-E2 contains 11 *N*-linked glycan residues, two of which are located within epitope I (Asn-417, Asn-423), one is present in epitope II (Asn-448), and two are located within epitope III (Asn-532, Asn-540). However, because these glycosylation sites are known to shift and modulate virus neutralization, it was decided not to consider these in the epitope design [40]. The rationale being that by keeping the design as simple as possible, there would be a higher likelihood of inducing a neutralizing antibody response that would be resistant to escape. Nevertheless, Pierce et al. [41] successfully elicited a neutralizing anti-HCV E2 antibody response while using a mono-glycosylated cyclic peptide that was based on epitope I. Sandomenico et al. [42] further illustrated this possible importance, who used a non-glycosylated cyclic peptide based on epitope I that did not result in a neutralizing anti-HCV E2 antibody response. Thus, introducing *N-*linked glycan residues could be required for a more accurate design of epitope mimics. However, **20**-mcKLH and **22**-mcKLH both elicited anti-HCV-E2 antibody responses without the incorporation of *N*-linked glycan residues. In turn, this suggests that there is a degree of ambiguity between the importance of various *N*-linked glycan residues. Still, the antibody response for epitope I may be increased by including the glycosylation sites within the peptide sequence. In addition, the glycosylation sites within epitope II might be even more important to stimulate a proper antibody response.

The design of the complete discontinuous epitope mimic **27** was highly ambitious in order to obtain a biomolecular construct, which incorporates all of the different substructures of a larger immunogenic region of HCV-E2. Multiple neutralizing antibodies have been isolated capable of binding residues that spread across epitopes I, II, and III. [26] Despite the considerable complexity of the design of discontinuous epitope mimic **27**, its mimicry could not be validated in vitro, as was achieved by ELISA for continuous epitope mimics **20**, **21**, and **22**. [32,33] The reason for this was that discontinuous epitope mimic **27** could only be obtained with an amine handle for conjugation, which was incompatible with the previously developed method utilizing maleimide-activated plates for ELISA [32,33]. In addition, discontinuous epitope mimic **27** contains three free amine functionalities that may prevent the selective conjugation in a single orientation, possibly also leading to cross-linking. 

In addition to above mentioned approaches for improving the design of discontinuous epitope mimics, further optimization could be achieved by the synthesis and testing of more variations of the conjugated cyclized peptides **4**, **5**, and **6** onto TAC-scaffold **19** (i.e., **4**-**5**-**6**, **5**-**6**-**4**, **5**-**4**-**6**, etc.; Figure 7).

## 4. Conclusions

In conclusion, we have been able to construct a complex fully synthetic discontinuous epitope mimic **27** that us based on the known antibody binding epitopes within the HCV-E2 glycoprotein using a modified orthogonally protected trialkyne TAC-scaffold **19**. This biomolecular construct included a flexible linker that was used to conjugate it to mcKLH through amide bond formation. In addition, we have re-synthesized our previously developed continuous epitope mimics **20**, **21**, and **22**, and attached these to mcKLH via thiol-maleimide conjugation. Both continuous and discontinuous epitope mimics were subjected to two independent immunization experiments to try and obtain anti-HCV-E2 glycoprotein antibodies. In this way, we have investigated the validity our epitope mimics as potential synthetic vaccines. 

Although epitope mimics **20** and **22** stimulated an antibody response that was capable of binding HCV E2 glycoprotein, unfortunately the purified IgG from the obtained sera were not able to prevent infection and, thus, neutralize HCVpp. An explanation for this failure may be a sub-optimal arrangement of the individual linear epitopes onto the scaffold. Perhaps the mimicry of HCV-E2 by the discontinuous epitope has to be improved guided by structural investigations of the molecular constructs. Further optimization of the individual epitopes, including the preparation of glycosylated derivatives and introduction of structural constraints, are other considerations. A promising avenue to pursue might be the use of a more pre-organized scaffold, as we have recently showed in the preparation of (synthetic) antibody mimics [29,30].

Nevertheless, the concept of epitope mimicry is still of great value for alternative—completely chemosynthetic—vaccine design. Still, a lot of work has to be carried out to validate the proof of principle of discontinuous epitope mimicry beyond our shown successful individual epitope mimicry [32,33]. This should be explored in a broader fashion with respect to peptide loop length, order of conjugation of cyclic peptides, sequence optimization, scaffold screening, and initial *in vitro* validation using ELISA with known isolated neutralizing antibodies. If ultimately successful, the strategy that is described here can be readily adopted to the design and synthesis of synthetic vaccines of a wide variety of viruses and other pathogens.

## Data Availability

The data supporting the findings are available within the main text. The raw data are available from the corresponding authors upon reasonable request.

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
