# Peer review of "Design and Synthesis of HCV-E2 Glycoprotein Epitope Mimics in Molecular Construction of Potential Synthetic Vaccines"

_viruses, 2021, doi:10.3390/v13020326_

Round 1
Reviewer 1 Report
Until now, there are no effective HCV vaccines. In this article, the authors tried to develop a chemistry-based approach towards fully synthetic peptide-based vaccines using epitope mimicry, by focusing on highly effective and conserved amino acid sequences in HCV. Although the results are not successful, I think the concept the authors demonstrated has some potential for future vaccine development. However, now, these preliminary data are not qualified to be published in the scientific journal. Future more solid data against anti-HCV activity are required to be published.
Author Response
Reviewer 1
Comments and Suggestions for Authors
Until now, there are no effective HCV vaccines. In this article, the authors tried to develop a chemistry-based approach towards fully synthetic peptide-based vaccines using epitope mimicry, by focusing on highly effective and conserved amino acid sequences in HCV. Although the results are not successful, I think the concept the authors demonstrated has some potential for future vaccine development. However, now, these preliminary data are not qualified to be published in the scientific journal. Future more solid data against anti-HCV activity are required to be published.
Response: We appreciate the comments of the reviewer. However, the data we have presented are not preliminary. Nevertheless, we do agree that there is reason to repeat the experiments also taking into account the suggestions, which we have delineated in the conclusions section.
Reviewer 2 Report
The authors have designed synthetic vaccine candidates based on HCV-E2 epitope mimics and tested both continuous epitope mimics and their combination forming a discontinuous epitope mimics for their capacity to elicit neutralizing anti-E2 responses in mice. Only one mimic based on the continuous epitope III induced substantial antibody responses that were not neutralizing, while another one (epitope I) induced only mild non-neutralizing antibody responses, and no antibodies specific for the discontinuous epitope were induced.
- In the material and method section, the design of the neutralization assay should be more detailed (volume of the inoculum, dilution?).
- The figure 5 is barely legible, which makes almost impossible the distinction between PB and FB groups. The panels A and B should be enlarged.
- The authors explain the possible lack of antibodies against the constructs 21 and 27 to the fact that the mimetic peptides were not glycosylated. Was the E2 protein used in the ELISA test glycosylated?
- Do the authors detect antibodies specific for the different epitope mimics in the serum from these mice (ie when coating ELISA plates with these mimics instead of the E2 protein), at least for the mimics 20, 21 and 22 as they previously reported? Perhaps the occurrence of antibodies against the different mimics, including the 27 could be tested by coating the ELISA plates with the KLH-coupled peptides.
- There are some typos to be corrected (eg line 766, Q, P, Q, and R instead of O, P, Q, and R).
Author Response
Reviewer 2
Comments and Suggestions for Authors
The authors have designed synthetic vaccine candidates based on HCV-E2 epitope mimics and tested both continuous epitope mimics and their combination forming a discontinuous epitope mimics for their capacity to elicit neutralizing anti-E2 responses in mice. Only one mimic based on the continuous epitope III induced substantial antibody responses that were not neutralizing, while another one (epitope I) induced only mild non-neutralizing antibody responses, and no antibodies specific for the discontinuous epitope were induced.
- In the material and method section, the design of the neutralization assay should be more detailed (volume of the inoculum, dilution?).
Response: As requested we have expanded the description of the neutralisation assay (lines 611-618). Neutralisation was performed using purified IgG rather than sera, this is to eradicate the possibility of effects due to other components present in the sera. We have also amended the legend for figure 6 to clarify this point.
- The figure 5 is barely legible, which makes almost impossible the distinction between PB and FB groups. The panels A and B should be enlarged.
Response: Although we think that the journal already has taken care of this, if required we will expand this figure.
- The authors explain the possible lack of antibodies against the constructs 21 and 27 to the fact that the mimetic peptides were not glycosylated. Was the E2 protein used in the ELISA test glycosylated?
Response: Yes the E2 protein used in the ELISA was glycosylated. The HCV E2 protein was expressed in mammalian cells and no modifications to glycosylation sites were introduced.
- Do the authors detect antibodies specific for the different epitope mimics in the serum from these mice (ie when coating ELISA plates with these mimics instead of the E2 protein), at least for the mimics 20, 21 and 22 as they previously reported? Perhaps the occurrence of antibodies against the different mimics, including the 27 could be tested by coating the ELISA plates with the KLH-coupled peptides.
Response: This is an interesting question, however the sera obtained from the mice was very limited therefore we had to be selective with the experiments we chose to perform.
Provided there was enough serum, we could have coated maleimide-activated plates with mimics 20, 21, and 22, like we have done before, and perform an ELISA using serum to verify elicitation of antibodies recognizing specific mimics. Of course, this also could have been done by coating ELISA plates with KLH-coupled mimics, including 27. This would have indeed provided us with insight in the general immunogenic potential of the mimics, but no information on the desired immunogenic potential against HCV. The limited amount of sera obtained forced us to prioritize experiments focussed on validating the immune response to HCV and not the mimics themselves.
- There are some typos to be corrected (eg line 766, Q, P, Q, and R instead of O, P, Q, and R).
Response: We have corrected this and we have checked the manuscript for other typos.
Response: we thank the reviewer for his/her comments and suggestions. Please see our responses in between the comments of the reviewer
Reviewer 3 Report
The article by Meuleman et al. studies the synthesis of hepatitis C virus (HCV) E2 glycoprotein epitope mimicry and its potential use as synthetic vaccines. The authors on cyclic synthetic peptides based on three epitopes being fragments of E2 proteins 411-424, 436-448, 521-537, respectively. As a scaffold, an orthogonally protected click-chemistry method has been applied (trialkyne triazocyclophane (TAC) Cu(I) catalyzed alkyne-azide cycloaddition (CuAAC)). The TAC scaffold has been further modified by introducing a linker for thiol-maleimide conjugation to a compatible carrier protein- mariculture keyholelimpet hemocyanin (mcKHL). All synthetic procedures and compound characteristics have been thoroughly described by the authors. Some of the discontinuous epitopes (numbered 20 and 22) exhibited binding to a soluble E2 (sE2) protein that was validated by ELISA. The immunized mice showed an antibody response, however the obtained sera were not able to prevent HCV infection. Although the prevention has not been observed, the work performed by the authors is important in the field of epitope mimicry. In the opinion of the reviewer, the article should be published in Viruses.
Author Response
Reviewer 3
Comments and Suggestions for Authors
The article by Meuleman et al. studies the synthesis of hepatitis C virus (HCV) E2 glycoprotein epitope mimicry and its potential use as synthetic vaccines. The authors on cyclic synthetic peptides based on three epitopes being fragments of E2 proteins 411-424, 436-448, 521-537, respectively. As a scaffold, an orthogonally protected click-chemistry method has been applied (trialkyne triazocyclophane (TAC) Cu(I) catalyzed alkyne-azide cycloaddition (CuAAC)). The TAC scaffold has been further modified by introducing a linker for thiol-maleimide conjugation to a compatible carrier protein- mariculture keyholelimpet hemocyanin (mcKHL). All synthetic procedures and compound characteristics have been thoroughly described by the authors. Some of the discontinuous epitopes (numbered 20 and 22) exhibited binding to a soluble E2 (sE2) protein that was validated by ELISA. The immunized mice showed an antibody response, however the obtained sera were not able to prevent HCV infection. Although the prevention has not been observed, the work performed by the authors is important in the field of epitope mimicry. In the opinion of the reviewer, the article should be published in Viruses.
Response: The positive comments of this reviewer are highly appreciated.